# Nurse-Led Bereavement Support During the Time of Hospital Visiting Restrictions Imposed by the COVID-19 Pandemic—A Qualitative Study of Family Members’ Experiences

**DOI:** 10.3390/nursrep15070254

**Published:** 2025-07-14

**Authors:** Michele Villa, Annunziata Palermo, Dora Gallo Montemarano, Michela Bottega, Paula Deelen, Paola Rusca Grassellini, Stefano Bernasconi, Tiziano Cassina

**Affiliations:** 1Department of Intensive Care, Cardiocentro Ticino Institute, Ente Ospedaliero Cantonale, Via Tesserete 48, 6900 Lugano, Switzerland; annunziata.palermo@eoc.ch (A.P.); dora.gallomontemarano@eoc.ch (D.G.M.); paola.ruscagrassellini@eoc.ch (P.R.G.); tiziano.cassina@eoc.ch (T.C.); 2Chief Nurse’s Office, Department of Healthcare Professions, Azienda Unita Locale Socio Sanitaria (AULSS) 2 Marca Trevigiana, Via Sant’Ambrogio di Fiera 37, 31100 Treviso, Italy; michela.bottega@aulss2.veneto.it; 3Chief Nurse’s Office, Cardiocentro Ticino Institute, Ente Ospedaliero Cantonale, Via Tesserete 48, 6900 Lugano, Switzerland; stefano.bernasconi@eoc.ch

**Keywords:** bereaved, intensive care unit, qualitative research, follow-up, COVID-19, family

## Abstract

**Objectives**: This study aims to explore the experiences of bereaved family members during and after the loss of a relative in an intensive care unit (ICU) during the COVID-19 pandemic-related visitation restrictions, as well as to assess their perceptions of a nurse-led bereavement support programme. **Methods**: Ten participants with a relative who had died in an ICU were recruited in September 2020 during a follow-up bereavement meeting at a tertiary cardiac centre in Switzerland. Descriptive qualitative research was conducted. Face-to-face nurse-led follow-up bereavement meetings, adapted to the pandemic circumstances and conducted as semi-structured interviews, were analysed by a thematic analysis. **Findings**: Fifteen sub-themes and three main categories were identified. The motivation behind the family members’ participation in the meetings was to ask and learn about their experiences regarding the death of their relative during this abnormal time. The reactions to the meetings varied among the families. Many expressed that the experience of bereavement was particularly challenging and painful, and that the absence of a final farewell to their loved one, as well as the impossibility of having a formally held funeral, made the deaths harder to accept. The families appreciated the interview as it gave them clarification, information, and an awareness of the facts and the care provided, and for several of them it was also a chance to share their emotions and express any difficulties they might have encountered both during and after the patient’s death. **Conclusion**: The COVID-19 pandemic’s restrictions had a profound impact on families who lost a loved one in an ICU. The nurse-led bereavement support service responded to the needs of grieving families, providing valuable emotional and practical support and re-establishing a healthy relationship between the families and the caregivers that was hindered by pandemic restrictions. The study also shows that a nurse-led bereavement support service can be a valuable component of family-centred care.

## 1. Introduction

During the COVID-19 pandemic, patients often died alone and far from their loved ones in hospitals, both of SARS-Cov-2 and other diseases. Due to the visitation restrictions that were imposed to control the spread of infection in healthcare settings, families experienced increased frustration and trauma at the loss of a loved one compared to so-called “normal” situations [1,2,3]. The inability to be present, particularly during a patient’s final moments, led to additional psychological distress for family members. This distress was further exacerbated by the impossibility of holding funerals attended by family and friends [1,4]. These constraints prompted widespread discussions on the emotional and psychological distress and ethical implications of these measures [5,6]. In this context, communication between healthcare providers and families was significantly altered, resulting in a decline in the quality of information conveyed, as well as a diminished sense of emotional support and connectedness [7].

Under usual circumstances, numerous intensive care unit (ICU) family support programmes are available for those who experience the loss of a loved one in critical care. Several models described in the literature outline a structured bereavement care pathway with multiple stages of follow-up, primarily consisting of information provision, condolence interventions, telephone calls, and family meetings [8]. Post-death follow-up meetings are conferences between the ICU team and the patient’s family following a patient’s death [9,10]. This service provides an opportunity for family members to consider the events, ask questions concerning the causes and circumstances of death, share their experience of grief, and receive reassurance about the appropriateness of the distressing symptoms often reported during this stage [11,12,13]. In general, ICU bereavement support programmes are predominantly physician-driven, either during the dying and death period or during follow-up meetings [10]. In non-emergency conditions, these pathway bereavement support services are generally appreciated and valued by families, although they are not widely used [8,14]. Moreover, this support is recognised as a key component of end-of-life care in ICUs, but it remains under-investigated, and the available literature provides only a low level of evidence. A recent randomised controlled trial failed to demonstrate that a Caregiver Pathway Intervention had any significant effect on the Post-Traumatic Stress Disorder (PTSD) symptoms of the family caregivers of non-surviving patients [15]. However, in certain healthcare settings, especially during the emergency phase of the COVID-19 pandemic, these services proved to be highly relevant and appreciated [16,17].

Following the first wave of the pandemic, a nurse-led bereavement support service for families who had experienced the loss of their loved ones in the ICU was promptly reactivated in the hospital after the first weeks of initial shock. While numerous studies have explored the experiences of families who lost relatives during the COVID-19 pandemic, only a limited number have examined the role and impact of nurse-led bereavement support services in this context [6,16]. Notably, few studies have reported on bereavement support services that were not purpose-built for research or temporary emergency implementation, but rather integrated into routine clinical practice and delivered by members of the care team. Therefore, it is important to explore the issues addressed in these follow-up meetings, as well as the experiences, expectations, and perceptions regarding the usefulness of this service. The results may serve as a resource for reflection, as well as to improve the best practices for engaging with bereaved families, and contribute to tangible improvements in clinical practice and bereavement support in future emergency situations.

## 2. Materials and Methods

### 2.1. The Aim of the Study

The aim of this study was to explore the experiences of bereaved family members during and after the loss of a relative in an intensive care unit (ICU) during the time when visitation restrictions were imposed by the COVID-19 pandemic. In response to the participants’ experiences of their losses, the study aimed to investigate their perceptions of a nurse-led bereavement support programme.

### 2.2. Study Design

A descriptive qualitative research method with a thematic analysis was performed. The Consolidated Criteria for Reporting Qualitative Research (COREQ) was used to report the results [18].

### 2.3. Context

The study was conducted at the Cardiocentro Ticino Institute (Lugano, Switzerland). The hospital is a cardiological and cardiac surgery hub of Canton Ticino, in southern Switzerland, serving a population of 350,000. It has nine ordinary intensive care unit (ICU) beds, which were increased by a further six ICU beds during the COVID period. In the first lock-down period, from March to May 2020, ‘hospital visiting hours’ were banned, so communication with family members or caregivers was restricted to telephone and/or video calls between the physician in charge and the caregiver.

### 2.4. Bereavement Support Service

Despite the pandemic, the nursing team promptly provided family members with support following the death of their relatives through a structured three-stage bereavement follow-up support programme (Figure 1), which has been active for over ten years [9].

Firstly, approximately four to five weeks after the event, a condolence letter signed by the ICU head nurse was sent to the family member identified in the medical record as the ‘contact person’. The letter expressed closeness to the family, acknowledged the difficult circumstances they had experienced or might still be experiencing, and informed them of an upcoming phone call from the ICU bereavement support service (Appendix A). Secondly, four weeks later, the ICU bereavement support service nurse contacted the caregiver by phone to offer condolences and support, and to answer any questions the family may have had about the care provided during their relative’s stay in the ICU. During the phone call, the nurse extended an invitation to a face-to-face meeting with the ICU clinical team. Thirdly, in September 2020, as soon as pandemic restrictions allowed, a follow-up bereavement meeting was planned. The aim of this meeting was to address the relatives’ questions regarding the patients’ ICU permanence and the circumstances surrounding their deaths. These meetings also provided a structured opportunity for family members to express their emotions, to receive emotional support, and to seek specialised assistance if they encountered difficulties in coping with the grieving process. During the meetings, the nurses also discussed the care provided to the family members’ loved ones during the terminal phase in the ICU. The meetings were conducted by a nurse from the bereavement support service and a physician, both from the ICU, with the intervention of social services if needed. The meetings were prepared for by the researchers (A.P. and D.G.M.) in collaboration with the physician, through a review of each patient’s clinical documentation and discussions with the ICU care team to address any specific issues.

### 2.5. Sample

Purposive sampling was considered. At the beginning of the follow-up bereavement meeting, the family members were introduced to the possibility of participating in the research, with an explanation of the purpose of the study and the need to provide their informed consent. Participation was offered to all family members attending the meeting—excluding those under 18. The sample included the relatives of people who died in the hospital between March and April 2020, a period during which relatives were banned from visiting during hospital hours. To capture a wide range of perspectives, a minimum of six participants was set as the recruitment target. Enrolment was closed upon reaching thematic saturation or exhausting the pool of eligible participants.

### 2.6. Data Collection

Two nurses (A.P. and D.G.M.)—experts in follow-up bereavement support and qualitative research methods—conducted the interviews in September 2020. Both are ICU nurses: A.P. was trained in “Counselling” and D.G.M. holds a degree in Palliative Care. Both interviewers worked part-time as team members in the intensive care unit where the research was conducted. The meetings gave the caregivers the opportunity to ask questions about their relatives’ ICU stays and the circumstances of their deaths, and to express their feelings and emotions, including any challenges they may have faced during the grieving process. Following the first part of the meeting, which focused on bereavement support and achieving the goals of the support service, the dialogue with the family members continued with a flexible semi-structured interview with open-ended questions designed to address the specific objectives of the study (Appendix B—Interview guide). All the interviews were audio-recorded and transcribed verbatim. The non-verbal components were recorded through handwritten notes to enhance the understanding of the emotions and feelings [19]. All the meetings were held inside the hospital, in a room located on a different floor, separate from the ICU, to avoid additional stress for the relatives.

### 2.7. Data Analysis

A thematic analysis of the audio-recorded and verbatim transcribed interviews was conducted by A.P., D.G.M., and M.V. following Braun and Clarke’s six-step process [20]: (a) familiarisation with the data; (b) initial code generation; (c) a theme search based on the initial codes; (d) theme review; (e) theme definition and labelling; and (f) report writing. The transcripts were read multiple times to grasp the participants’ experiences. The key phrases describing these experiences were identified and assigned codes. Two researchers (A.P. and D.G.M.) independently generated the initial codes manually, using tabular formats. The codes with similar meanings were then merged and organised into conceptual themes and sub-themes collaboratively by the same two researchers, together with a third researcher with expertise in qualitative research (M.V.). Representative quotes illustrating each theme and sub-theme were also selected as part of the analysis process and then translated into English by a native speaker. To enhance the reliability of the analysis and to support the development of the codes and themes, a qualitative research expert with a PhD (M.B.) reviewed the entire analytic process. After including all the participants available for recruitment, the researchers concluded that data saturation was reached during the seventh meeting, with ten participants.

### 2.8. Ethical Considerations

The study was approved by the Ethics Committee of Canton Ticino (Switzerland), approval number 2020-01916/CE 3718. Written informed consent was obtained before starting the interviews. To protect the participants’ confidentiality, the interviews were anonymised and coded with an alphanumeric code. Moreover, in cases of significant psychological trauma, the possibility of a session with a hospital psychologist was also offered.

### 2.9. Accuracy and Trustworthiness

During the meetings, no feedback or negative or positive reinforcement was given regarding the considerations and/or opinions about the helpfulness of the service in order to ensure the participants’ feelings of freedom to share their experiences [21]. The meetings were transcribed in full by the interviewers (A.P. or D.G.M.) within a few days, ensuring their accuracy and conformity. Due to the COVID-19 restrictions, it was inappropriate to reconvene the participants to revise and/or receive feedback on the transcripts. Furthermore, in order to ensure the accuracy and trustworthiness of the findings, direct quotes were extracted, and several briefings (including A.P., D.G.M., and M.V.) were performed to discuss the emerging codes and themes until a consensus was achieved [22]. Transferability was sought through the provision of details on the setting and context.

## 3. Findings

Twelve families were contacted to participate in the bereavement support follow-up meetings. Of these, seven agreed to participate, while four declined as they did not see any benefits in doing so. One family declined as they did not wish to relive the distress of their relative’s death. All the families who attended the follow-up meetings agreed to participate in the study. A total of ten participants were interviewed, with a median age of 43 years (1st and 3rd quartile, 34–60), the majority of whom were women (8/10). The median age of the deceased patients, on the other hand, was 77 years (1st and 3rd quartile, 60–84), and they were predominantly men (3/7). Three interviews were attended by two family members, in two cases a daughter and a spouse, and in one case, a son and a daughter. One interview was attended by a nephew, while the remaining three participants were attended by two daughters and one spouse (Figure 2).

The median interview duration was 68 min (1st and 3rd quartile, 57–74), with a maximum duration of 92 and a minimum of 48 min. Two interviews were held on a Saturday and Sunday morning on the request of the participants, while all the other meetings took place during the week in the early afternoon. No serious symptoms of psychological distress requiring additional support from a psychologist, or the intervention of a social support service, were identified during the interviews.

Three main themes were identified: (i) the relatives’ need to understand what happened; (ii) the factors influencing the grief processing journey; and (iii) the evaluation of the meeting (Figure 3). Exemplary quotes from the interviews are presented in Table 1.

### 3.1. Theme 1: The Relatives’ Need to Understand What Happened

The motives that encouraged family members to participate in the follow-up meeting were the desire to be informed and aware of the events, including the causes, the diseases, and possible prediction and/or anticipation of the causes of illness and death of their relatives. Specific requests were connected to the final moments before death. During the meetings, the family members first enquired about the disease and the causes of death, asking for clarifications of the specific clinical circumstances. In several cases, the caregivers struggled to fully understand the information provided by the physicians during hospitalisation and also misinterpreted some of the information. As a result, they expressed the need to clear up doubts and to clarify inconsistencies in the information received (Need to fill gaps about events, #Q1–3). The conversation also became a time for clarification, recriminations, and the venting of feelings that developed in the months following the deaths (Recriminations, #Q4, #Q5). The requests for a reconstruction of the final hours of their relatives’ lives and the need for information about the care and assistance provided at this time were also clearly a strong and emotional need (Finding out about the last hours of life, #Q6–8). The caregivers expressed a strong desire to receive clear and comforting information about the circumstances of their relatives’ deaths, especially regarding the patient’s level of awareness and comfort in their final moments. Learning about a death without pain instilled a much-needed sense of relief (Need to be reassured, #Q9–11). The meetings also allowed the participants to express their profound regrets, such as not being able to see their loved ones at the time of death—a missed opportunity that left them with a lasting emotional burden (Remorse and regrets, #Q12–14).

The restriction on hospital visits due to the COVID-19 pandemic had a strong impact on the experiences of family members, as it denied them the opportunity to see their loved ones during their final moments. This problem was exacerbated even further in situations where rapid clinical developments did not allow for any visits and/or video calls whatsoever (Need to be present, #Q15–17). A fear of the contagion was one of the presumed reasons for delays in going to the hospital and/or seeking help (Delayed request for help, #Q18, #Q19).

### 3.2. Theme 2: Factors Influencing the Grief Processing Journey

The COVID-19 restrictions had a significant influence on the participants’ experiences, causing heightened concerns and fears. One of the greatest challenges expressed by the relatives was the inability to be at the patient’s side. In addition, in several cases, these restrictions limited and complicated funeral rituals, preventing family members from sharing their grief and pain with their loved ones and friends (Modified funeral rituals, #Q20, #Q21).

During the conversations, the pain of losing a loved one rapidly came to the surface. For all the participants, accepting grief was described as an ongoing, arduous journey, characterised by different levels of awareness, resilience, resources, and strategies for coping with bereavement. The follow-up bereavement meetings were often perceived as a moment to vent feelings and emotions, and crying episodes were frequent. On the one hand, crying revealed that the pain was still very much present and, on the other hand, it had an important liberating and propaedeutic function in the grieving process.

The participants sincerely and spontaneously expressed emotions related to the loss of their loved ones, such as feelings of powerlessness, remorse, suffering, frustration, inadequacy, incredulity, disbelief, and loneliness (Difficult emotions, #Q22–24). In coping with the pain of loss, in order to overcome the difficult moments of bereavement and return to normal life, the respondents revealed different strategies depending on their inner resources and family and social backgrounds (Resources and Coping Strategies, #Q25–28). The experience of bereavement was not just a personal issue, and often involved interactions with other family members, including their reactions, behaviours, and difficulties (Family grief processing, #Q29, #Q30). The patient, as a subject, was often spontaneously put at the centre of the narrative, with his/her life stories from the beginning of the journey through illness and suffering. Retelling, remembering, and explaining even complex family dynamics were a necessary source of relief (Memories and stories of a life spent together, #Q31–34).

### 3.3. Theme 3: Evaluation of the Meeting

Within the families, the expectations of and considerations on whether to attend the meeting varied. Several family members did not want to join as they feared that painful feelings might be revived, while others did not take part as they had misgivings about the real necessity and purpose of such meetings. On the other hand, others accepted the invitation to attend as they wished to gain information and clarification on the events and circumstances surrounding the death of their relatives (Pre-interview concerns and expectations, #Q35–38). At the end of the meetings, the overall evaluations were positive, and the service was highly valued by the families. It was seen as a useful opportunity to clarify and learn the facts, as a means of providing reassurance and awareness of reality, and as a way of sharing the feelings they experienced during the months following their relatives’ deaths. In all the interviews, the family members expressed appraisal, satisfaction, and gratitude for such an opportunity during their time of grief (Acknowledgements, #Q39–41). At the end of the interviews, the participants frequently reported increased awareness following the explanation of the causes and circumstances of death and the care provided to their loved ones (Increased awareness, #Q42, #Q43).

## 4. Discussion

To the best of our knowledge, this is the first study reporting the experiences of those receiving bereavement support service offered by nurses during the first wave of COVID-19, in the socio-cultural context of central and southern Europe. In the interviews, the participants openly reported their experiences of distress and suffering and felt free to share their feedback and observations. Their requests for clarification about their relatives’ illnesses and final moments, as well as their openness in expressing their innermost feelings, showed both the importance of these meetings and the strong feelings provoked by such an experience.

Based on previous experience, an effort was made to offer families the opportunity to take stock of past events, to learn more regarding the causes and circumstances of the deaths, to share their suffering, and to meet the people who took care of and stood by their loved ones. To achieve this objective, slight modifications were made to the structure of the bereavement support service by including a physician from the outset. This strategy proved to be appropriate and successful, as all the participants expressed a strong need to inquire about their loved one’s illness and death through specific questions, which were thoroughly addressed by the physicians. This finding aligns with previous reports by Kock and colleagues, where 91% of family members during pre-pandemic follow-up meetings post-death felt the need to discuss the causes of death, and 61% required the presence of physicians [13].

Difficulties in dealing with the suffering related to the experience of the hospitalisation of a relative were still present during the interview. Separation from loved ones during their final moments, as well as the isolation and the difficulties in social relationships during the pandemic, complicated and sometimes impeded, the physiological process of bereavement [23]. Most family members had questions about the illnesses and deaths of their loved ones, and the answers to such questions were necessary in order to connect the events and obtain an overall clearer picture [11], as an important step in the grieving process.

The nurse-led meetings provided a supportive environment that facilitated sharing and helped the family members to feel welcome and understood. The value of these interviews was evident not only in the families’ requests for information and clarification, but also in their ability to verbalise their experiences. Additionally, these face-to-face meetings offered reassurance about the normality of their emotional responses during different stages of the grieving process. This initiative aligns with the report by Jeitziner and colleagues, which highlighted the importance of a family-centred, end-of-life care model, namely the role of a structured bereavement support service that integrates the expertise of a range of healthcare professionals to enhance the emotional support and communication for grieving families [24,25]. Nurses play a pivotal role, as their position places them in close contact with both patients and their families. This enables them to provide a unique combination of continuity, presence, and humanisation of care during and after a critical illness, which makes them well-suited to lead bereavement support programmes [26].

Even in pre-pandemic times, unsatisfactory interactions between family members and clinicians and the inability to be close to a patient during the end-of-life stage were associated with prolonged feelings of prostration, generating a complicated grief process [6,27]. In fact, one of the frequent issues that arose was the need to know whether their relative had suffered from loneliness, whether he/she had displayed signs of feeling abandoned, and whether he/she had asked for his/her family members. These uncertainties often resulted in emotional distress, ranging from anger to guilt, and difficulty in achieving psychological closure with what had happened [28].

The pandemic has reaffirmed the importance of family-centred care, particularly in ICUs [29]. Several reports have described the care interventions implemented in ICUs during the pandemic to support patients’ families during the end-of-life phase of their loved ones [30,31]. In contrast, bereavement support services have been under-reported, probably due to their limited implementation even in the pre-pandemic era, as indicated by various national surveys [14,32,33,34]. Moreover, to our knowledge the generic bereavement support offered to families during the COVID pandemic, when available, was predominantly provided by psychologists or other professionals outside the care team [16]. We argue that support provided by the intensive care team enhances opportunities to address significant family concerns, improves the understanding of events, offers emotional support, and provides non-judgmental validation of the broad range of emotions that may arise in these circumstances.

The receipt of the condolence letter also elicited emotional responses. Although recent recommendations from the European Society of Intensive Care Medicine advise against routinely sending condolence letters written by ICU teams to bereaved family members [35], we believe that, within a structured bereavement support programme, the letter represents one of the elements for re-establishing contact with a family after a patient’s death. In this context, it should not be regarded as a standalone intervention capable of influencing grief symptoms, but rather as a component of a broader programme. As observed with some participants, the letter may nonetheless evoke emotional reactions that can be explored and addressed during subsequent follow-up encounters.

It is also noteworthy that the expectations and prejudices of the family members regarding the meeting proposed by the service were ambivalent. Within the families themselves, the proposed meeting ranged from instilling feelings of gratitude for an unforeseen opportunity to a lack of interest in an initiative deemed intrusive and almost irreverent. A common reaction within this range of legitimate responses was the feeling of surprise at being put in the spotlight.

### Limitations

The study has several limitations. Firstly, it was conducted exclusively in a single hospital in the Italian-speaking region of Switzerland, where the resilience of the healthcare system to the first pandemic wave differed from other contexts. Nevertheless, it should be noted that the containment policies were the same as those adopted in the highest-income European countries. We therefore believe that the issues addressed in the meetings and the considerations that emerged can help to understand the value and potential of bereavement support in other contexts. Secondly, the number of participants was limited to ten family members across seven ICU deaths due to the exhausting of the pool of eligible participants. However, we believe that valuable insights have emerged that can contribute to the aims of our study and are therefore worth sharing. Thirdly, the cardiac ICU setting is not comparable to the hub of COVID ICUs that faced the bulk of the pandemic on the front line. Nevertheless, as pointed out by Jeitziner and colleagues [24], non-COVID patients were still admitted to intensive care units, and it is important that these patients and their families are not considered any differently. These relatives of deceased patients also experienced a severe grief reaction with intense yearning or separation distress due to their bereavement [24]. Fourthly, in several meetings the presence of multiple family members may have limited individual freedom of expression regarding personal experiences and perspectives. Finally, we acknowledge that reconstructing ICU events during the meeting, held after a period of time, was particularly challenging, especially given the complexity of care during the COVID-19 pandemic. Nevertheless, these difficulties were mitigated by the expertise of the bereavement support team and the availability of detailed nursing documentation, which allowed for sensitive and accurate responses to family members’ questions, even months after a patient’s death.

## 5. Conclusions

During the meetings the bereaved family members described a distressing experience, both during the hospitalisation and after the patient’s death. The families requested a reconstruction of the facts, as faithfully as possible, without superficial minimisation, viewing it as a fundamental substratum to understand a reality otherwise only imagined and hypothesised. The families showed their appreciation for the bereavement support service, considering it to provide valuable emotional and practical support in understanding what they were denied and a way to process the experience of losing their loved ones during the harsh events of the pandemic.

The profound impact on family members illustrates the need for visiting policies during pandemics to be reassessed, along with the development of specific ICU family-centred care frameworks for future crisis management. The study also shows that nurse-led bereavement support can be a valuable component of family-centred ICU care. Finally, integrating this service into routine clinical practice could also optimise professional roles and enhance skills for managing emergency scenarios.

## Figures and Tables

**Figure 1 nursrep-15-00254-f001:**
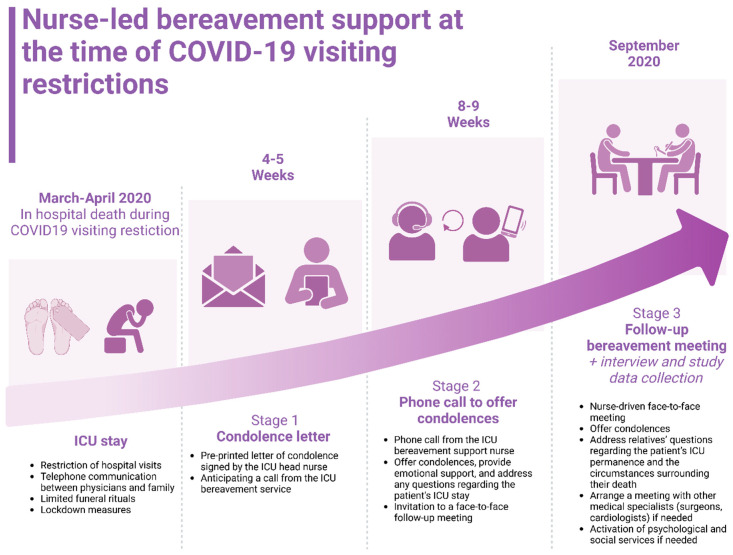
Nurse-lead bereavement support framework.

**Figure 2 nursrep-15-00254-f002:**
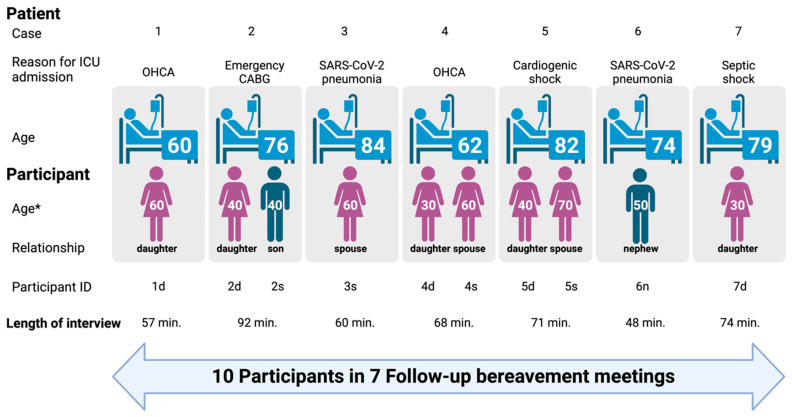
Characteristics of the study participants. * To protect their privacy and confidentiality, the age of the participants is expressed in decades. ICU—intensive care unit; OHCA—out-of-hospital cardiac arrest; CABG—coronary artery bypass graft; SARS-CoV-2—severe acute respiratory syndrome coronavirus 2.

**Figure 3 nursrep-15-00254-f003:**
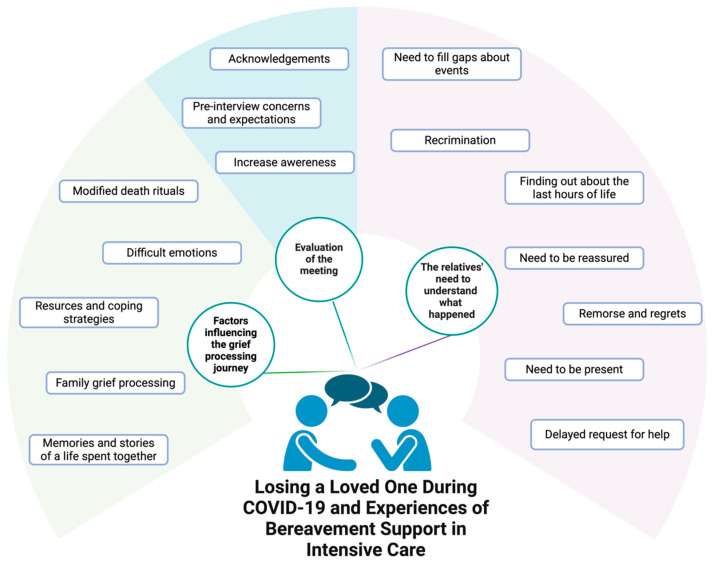
Identified themes and sub-themes from thematic analysis.

**Table 1 nursrep-15-00254-t001:** Themes, sub-themes, and exemplary quotes from the interviews.

Main Theme 1: The Relatives’ Need to Understand What Happened		
Sub-Themes	#	Quotes (ID Participant)
Need to fill gaps about events	1	*I just want to know what happened the day he got here. All I know was that one moment he was on the phone with his girlfriend and the next he was being intubated…* (7d)
	2	*We knew about his heart condition, but he lived a long life. It’s hard to make sense of it, all these thoughts that come up afterwards are hard to process.* (5d)
	3	*The first question is about re-capping everything… not that it’s not clear, by now everything’s clear. At the beginning you had administered her antibiotics because it was written that whoever had performed the heart massage had transmitted something to her. Not sure about this, I think I’m still wrapping my head around this part.* (1d)
Recriminations	4	*At home, my mother and I were wondering, because the matter is still somewhat… unclear. He had had a medical check-up done, after which no follow-up was done because of the COVID outbreak. And then he got admitted here as an urgent case… and it’s unclear if he should have maybe been admitted sooner…* (5s)
	5	*The night he died, I must say I was really angry with the nurse who asked that I call later as apparently they were busy with some paperwork at the time. A few hours later, my brother called to tell me that dad had passed away. I’d like to talk about this. I’d like to make it understood that this is a delicate matter and to be more sensitive because… when a father dies… and until the very end I couldn’t be with him or see him… I mean, knowing my father is there, sick, and me being completely cut off. Is it too much to ask to know something about it?* (2s)
Finding out about the last hours of life	6	*Ever since the ambulance took him away, we were unable to reach him… The following day at three o’clock they called, telling us that he’d died. I’d like to know if he was aware, if he said anything. Can it be possible that he didn’t utter even a single word?* (5s)
	7	*I’m unsure whether he was aware of what happened or not. If he really never asked for me, if prior to surgery he had asked to see someone or have something, in the end I don’t know, I mean he must have been confused, scared…* (2s)
	8	*I need to know if he passed away peacefully, and I know it’s a hard question to answer, but you know, my children are asking me if their uncle liked the drawings that we sent him…* (6n)
Need to be reassured	9	*Another question, because I wasn’t there unfortunately, that I’m being asked by her, and by my brother is: was he awake and aware? Was he sedated during his passing?* (2d)
	10	*Yes, what you told me about him passing away peacefully, unaware, has helped me. This is something that eases my mind a little… it helps dealing with the thought that we weren’t there during the darkest moments.* (5d)
	11	*It’s really nice to hear what you’re telling me, it helps me knowing that he passed away peacefully.* (4s)
Remorse and regrets	12	*Then there’s that thought that still lingers: maybe I could have, not seen him alive necessarily, but you know, be there for him. Being able to say that, even though he’d already made his decision, I could be close to him….* (2d)
	13	*I would have liked to see him just one more time. It was a wish of mine, one that I was denied. For my personal story it would have been an important moment. I was there in the funeral home, and it was there that I saw my father, alone, inside the coffin. I regret that he left this world alone, that he had none of us there with him.* (2s)
	14	*I asked myself this later, once I had realised, “why hadn’t I asked if it was possible for me to be there in the hospital”. Surely you would have told me it wasn’t, but it’s something that I now regret not having asked; it’s a burden I carry inside me, I feel the guilt.* (5d)
Need to be present	15	*... it shouldn’t have happened during this situation, this COVID situation; in the end, that night my mum called for an ambulance, and that was the last time we saw him. That is what is really, really hard to accept.* (4d)
	16	*I wanted to see him, but they kept telling me: “We can’t let you in”. I told them that I needed to see him, talk to him. “No, it’s not possible,” they insisted. I only saw him again at the funeral. He is dead…* (3s)
	17	*... I could have mustered up the courage to see him intubated, even while hooked up to a machine, for even a second. Even as he was dying. But I saw him in a coffin. I couldn’t do anything….* (2s)
Delayed request for help	18	*If it hadn’t been for COVID, he would have seen the medic in time.* (4s)
	19	*It was because of the Coronavirus situation that I told her: “mum, if you’re not feeling well, we can go to the hospital.” And she answered: “No, no, that’s where the Coronavirus is… She was terrorised by this COVID business, and she would never have called. She decided all on her own, she could have called anyone, but she didn’t.* (1d)
**Theme 2: Factors influencing the grief processing journey**		
Modified funeral rituals	20	*When he died. Eh. They called me on my phone telling me that Mr. X had died, but that I couldn’t come to the hospital. During the funeral I saw him inside a bag, inside a coffin. Only five people from the family were allowed to be present. But why? So many people knew him and would have liked to be present, but we were powerless.* (3s)
	21	*We were lucky enough to be able to arrange a small funeral, with about 10 people, out in the open. My father’s partner couldn’t come however, as she was out of Switzerland at that time.* (7d)
Difficult emotions	22	*When you then lose your last parent, you’re overcome by the feeling of being an orphan. That’s what it feels like. Doesn’t matter if you’re not a child anymore. When you lose your original family. This is what hurts most and is so hard to accept… and what’s more my brother’s far away.* (7d)
	23	*I don’t feel so well, I feel like crying now… because the day I lost him I refused to cry… he died here in the hospital. I still can’t bring myself to accept that he died. Today’s the first time I’ve allowed myself to cry. From the day he left I still hadn’t cried.* (3s)
	24	*... I want to remember her alive, still warm. And then we saw her, at the funeral home, inside the coffin… I wasn’t sure if I wanted to look inside. I told myself: “Come on! It’s my own mother!” (crying)… it’s not something so obvious for everyone… damn it! I hadn’t cried in two weeks.* (1d)
Resources and coping strategies	25	*Working has helped me. I spent a few days at home, but then started going in again. Working has allowed me to get out of my head. Once you’re at work, you leave other problems at the door. That has helped me, it’s been a useful resource.* (5d)
	26	*It’s just the two of us now, and we support and encourage each other.* (5s)
	27	*My dad’s being extremely supportive, and my partner and my friends are also helping. I have two good friends from the volleyball team who have also lost their mother, they give good advice and have gotten to be quite close.* (1d)
	28	*I’ve also spoken about this to the priest. We talked about family relationships and marital relationships. He knew him well, and has been really helpful through all of this.* (5s)
Family grief processing	29	*Still today his partner, despite the fact that it’s now been several months, has difficulties sleeping cause she sees him there. Lying on the floor. Compared to the first days, she now seems to be feeling slightly better.* (2s)
	30	*Before going out, my younger son asked me why I still come here… that evening he cried all night long. The older one is more introverted, closed off, and I think he still hasn’t processed the whole thing, he needs time. We didn’t take them to their grandpa’s funeral.* (2s)
Memories and stories of a life spent together	31	*He was my father’s cousin, so something like an uncle, but he really did a lot for us. He taught me many things, just like an uncle would, like tying my shoelaces when I was little. My children have always seen him as a grandfather of sorts. They never met their real grandpa. In the end, they did have a grandfather.* (6n)
	32	*A few months ago, it was me and my brother in one room, while my son was in the other room with grandpa and they were chatting. It was a rare and pleasant situation as my father never used to speak much. The pleasant memories need to slowly replace the painful ones.* (7d)
	33	*He didn’t live with us, he left home when I was 6 years old. It could be that my relationship with him improved when I got married.* (2d)
	34	*... we talked a bit about us, about our story.* (2s)
**Theme 3: Evaluation of the meeting**		
Pre-interview concerns and expectations	35	*When I got the letter, I immediately thought: “Wow what a nice service. It’s unusual to find hospitals offering something like it. How nice!” My mum’s partner, on the other hand, immediately dismissed it, saying that he wouldn’t come. He didn’t want to relive the whole experience. However, as soon as I got here, I said to myself: “Why did I to this, now everything’s going to come back to the surface”.* (1d)
	36	*My daughter was against me coming here. I told her that I wanted to do this because I wanted to know, because in any case thoughts and questions always come to mind about this. I want to know how he died, if he called for us, if he felt distressed. That’s how it is, isn’t it?* (5s)
	37	*After you called, I said to myself: “Let’s do it”. My husband told me that it would only hurt me, going back to the hospital means reliving the whole thing, experiencing it again. I don’t agree, I need to put the pieces together.* (5d)
	38	*My brother didn’t want to come because he was afraid he’d get angry.* (7d)
Acknowledgements	39	*I really appreciated this, thank you. I feel a bit better as I was finally able to talk about what I had bottled up inside. I’ve never cried until now. I couldn’t talk about it openly with my daughter because the whole thing makes her cry every time it comes up.* (3s)
	40	*We now have no further questions, talking about this has brought us some relief. Well done. It’s a nice thing you’re doing, really.* (5s)
	41	*I think that what you’re doing here can be truly helpful, more than ever in this moment. People need this.* (5d)
Increased awareness	42	*It’s been good for me as at least I know what really went on with my father. I wasn’t involved in the situation at the time, as I was informed mostly through what my brother told me. He would come to me and tell me: “Look, they decided this and that. and so on and so forth”. Thanks to you I now have had a genuine and direct testimony.* (2s)
	43	*I can’t say I’m at peace, because it’s really going to take a lot of time as we were very close, but still talking about it to you and learning a few more facts has helped me.* (5d)

## Data Availability

The data for this study can be made available upon reasonable request to the correspondent author (M.V.).

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
