# Peer review of "Nurse-Led Bereavement Support During the Time of Hospital Visiting Restrictions Imposed by the COVID-19 Pandemic—A Qualitative Study of Family Members’ Experiences"

_nursrep, 2025, doi:10.3390/nursrep15070254_

Round 1

Reviewer 1 Report

Comments and Suggestions for Authors

It was a pleasure to read this text. Right from the abstract. The authors piqued my interest with the quality of their writing.

The results are rich.

I really appreciated the figures and tables. Figure 2 in particular.  

The nurse support programme is very inspiring and innovative. This text is original and can support intensive care nurses, and certainly the population of bereaved people.

More concretely, here are a few constructive questions/comments:

First, it might be interesting to have details on the development of the program you evaluated: How, who, when? I understand that people were mobilized during the covid to develop the program ?

Could we have an example of the letter of condolence that was sent? I imagine it was personalised? who signed it? the nurses who were with the family?

Why wait 4 or 5 weeks to send the letter and call the family? were you inspired by a theoretical framework? data from other studies? This information is important because you evaluated a program.

I appreciate the interview guide!

How did you analyzed the interviews and choose the quotes ? How did you proceed to analyze ? we need more details for what you used a method to analyse.

You haves many themes and sub-themes, it's a shame you don't have more space to tell us more about them. However, the analyses are interesting and relevant. 

In the discussion, would it be relevant to specify how the face-to-face interviews were relevant and, above all, why they should be held by a nurse? There is an excellent opportunity here to talk about the nursing role and the added value of having nurses provide support to the bereaved.

Reviewer 2 Report

Comments and Suggestions for Authors

This is a worthy and interesting study that explores the experiences of bereaved families during the Covid pandemic who lost loved ones in ICU. As identified in the study this was an unusual and very difficult time for families having a profound impact. The study describes some interesting and valuable findings with regards the experiences of families losing a loved one at the time of the pandemic.

Introduction:

The is generally clearly written and provides a short but succinct overview of the very difficult family circumstances when experiencing a death in ICU. The seconds paragraph describes ICU bereavement services. I would have liked to also read here an evaluation of the effectiveness of such services. This would help to inform the gap in knowledge this study aims to fill. Line 69 mentions the reactivation of the services. What does this mean, had the normal service stopped and why? Line 73 identifies two studies that have already explored this issue, what is not address by these that this study aims to add? The final sentence gives some suggestion as to the implications for practice but as the reader I would like a stronger reason to continue reading.

AIMS (lines 80-85). The introduction and title identifies the need to explore the impact of the ICU bereavement service, which explained as the second study aim. There is nothing in the introduction to identify why there is a need to explore the experiences of the bereaved family in ICU during  the pandemic, which is the first study aim. Currently this reads as two studies.

2.5 SAMPLE Line 119-120 Is it correct that at the beginning of the interview participants(family members) were invited to take part. Surly if this is the beginning of the interview, research has already commenced? Some clarity regarding recruitment is needed.

2.6 Data collection. The researchers, as experts, clinical experience is highlighted but not their research experience.

3 Findings

PG5 very good informative infographic.

PG 6 again very good informative infographic

Themes 1 and 2 These are clearly presented and evidenced by table 1 and give the reader a good insights into the families experiences of the death of a loved one in ICU during the pandemic. Some powerful themes and emotions are uncovered. Interesting reading.

Theme 3 256-271. The discussion places a lot of emphasis on the evaluation of the meeting. The exemplars from table 1 don’t follow through to the text reporting on theme three, which reads more of about who accepted invitations to proceed than an evaluation of the service. As a result I would like to read more in the findings section about the evaluation of the service to give credibility to  the claim this is an evaluation of bereavement service and not a report on families experiences of the death of a loved one on ICU during the pandemic.  

4 DISCUSSION

Currently I read in the discussion how families experiences bereavement following the loss of a loved one in ICU during the pandemic. There is some connection to bereavement support but the main purpose of the study is to explain the family members experiences of the bereavement support service and I am not getting this. For example paragraph 3 (lines 292-299) describes the suffering due to separation families experienced, but does not explore the experiences of engaging with the service to overcome this.  Similarly paragraph 5 (lines 310-317) discusses clearly the worry of loneliness but what was the participants experience of engaging with the service to help with this? These are just examples and suggestions of where I think the bereavement service could be more present in the discussion section to make the article about family members experiences of bereavement support.

5 CONCLUSION

This gives a good direction for practice and highlights most of the key points. Due to the points above I am uncertain the study currently support the statement lines 360-362 “The families showed their appreciation towards the bereavement support service, considering it to provide valuable emotional and practical support in under standing what they were denied and a way to process the experience of losing their loved ones during the harsh events of the pandemic”

Reviewer 3 Report

Comments and Suggestions for Authors

Thank you for the opportunity to review this interesting study. Covid 19 left a mark on society around the world. Particularly traumatic was the isolation and fear of the unknown and the separation of family ties in times of hardship. The article in its attempt to explore the extent to which the constraints of the COVID-19 pandemic have impacted families who have lost a loved one in the ICU. The inability to be present, especially in the patient's final moments, led to additional psychological suffering for family members. Grieving family support was an extremely valuable part of nursing practice.

In the reviewer's opinion, the study should be supplemented with information on:

  1. the nursing bereavement support service was activated in the hospital mentioned in this study - by reading we do not know what kind of procedure it is and what elements of support are involved; informative, supportive should be supplemented;
  2. observations of the bereavement support service for families after death in the ICU, to what extent. Here a clinical psychologist plays a significant role.
  3. the purpose of the study was to explore the experiences of bereaved family members during and after the loss of a relative in the intensive care unit (ICU) during the pandemic - concerning what elements?
  4. to what extent the nurses responding during the meetings were able to provide information about the patient's stay and death in the ward, whether they were on duty, on the basis of what they gave answers related to the patient's stay;
  5. on the basis of what the interviewers provided information about the treatment process.
  6. The survey should focus on the expectations of families/caregivers for support in the period after the loss of a loved one;
  7. services highly rated by the family - which ones? Based on what criterion;
  8. no defined questions about feelings, emotions;
  9. Reconstruction of facts during the healing process and information about the death is done on an ongoing basis in direct contact with the doctor - to what extent is it possible to reconstruct the situation in distant time, when there is a lot and dynamic happening in the ICU;

Reviewer 4 Report

Comments and Suggestions for Authors

This is an excellent article demonstrating the need for patient centered care and also explored about the feelings of loved ones experienced during the loss of loved ones during the pandemic.  Yes, there are some limitations as this was at one facility so generalization to other locations is limited, however the emotions of the subjects in the study are most likely universal.  

Round 2

Reviewer 2 Report

Comments and Suggestions for Authors

REVIEWER COMMENTS:

The aim of the study is to explore Family members experiences of ICU nurse-led bereavement service after losing a loved one when visiting was restricted. As part of the evaluation family members have told their story of their experience at the time of the loss on ICU. For me the two are connected, but I worry that the reader may see the family experience as an addition rather than a cohesive story. I think this could be easily remedied with some careful wording of the aims section and possibly title, as suggested below.

1 Introduction:

Is clearly written and now presents a good overall justification for the study.

2.1 Materials and Methods

Aim of the study:

I read the aim of the study as two parts. What I still struggle with is the title as “Nurse Led Bereavement Support.. family members experiences” and the Aim of the study lines 88-92 being this AND family members experiences of ICU when losing a family member. To me there needs to be one story; how family members experienced loss in ICU and how effectively the nurse led bereavement services responded to those family’s experiences. Being blunt, the reader may interpret this as the researchers set out to evaluate the Nurse led bereavement service but added in asking about their experiences of loss. Rather than we wanted to know families’ experiences at this unique time and how effectively the bereavement service was able to respond. Perhaps:

“A qualitative study of family member’s experiences of a losing a loved one in ICU, at a time of Covid 19 Pandemic visiting restrictions and, in response, the evaluation of a nurse-led bereavement service” . Might be a bit wordy.

The aim of the study was “to explore the experiences of bereaved family members during and after the loss of a relative in an intensive care unit (ICU) at the time when visitation
restrictions were imposed by the COVID-19 pandemic. In response to participants experiences of such a loss the study aimed to investigate their perceptions of the nurse-led bereavement support programme”.  

Remainder of materials and methods now reads more clearly.

3 Findings.

3.3 Theme 3. I accept the researchers’ response.

4 Discussion: lines 332-335 additional section about nurse’s role, needs referencing.

5 Conclusion. I think the conclusion does a good job of pulling the threads together.     

Comments on the Quality of English Language

Clearly written

Reviewer 3 Report

Comments and Suggestions for Authors

In the reviewer's opinion, the authors did not provide comprehensive information on what nursing services were implemented to support bereavement. References to other publications do not approximate or imply.  The recommendation is to share the main elements of an essential task in a nurse's practice.
